# An Updated Checklist of the Genus *Capparis* L. (Capparaceae) in Vietnam, including a New Species from Hon Tre Island

**DOI:** 10.3390/plants11233402

**Published:** 2022-12-06

**Authors:** Silvio Fici, Leonid V. Averyanov, Danh Thuong Sy

**Affiliations:** 1Department of Agricultural, Food and Forest Sciences, University of Palermo, Viale delle Scienze ed. 4, 90128 Palermo, Italy; 2Komarov Botanical Institute of the Russian Academy of Science, Professor Popov Str. 2, 197376 St. Petersburg, Russia; av_leonid@binran.ru; 3Faculty of Biology, TNU-University of Education, 20 Luong Ngoc Quyen, Thai Nguyen City 250000, Vietnam; thuongsd@tnue.edu.vn

**Keywords:** *Capparis* sect. *Monostichocalyx*, climber, distribution, diversity, ecology, Indochina, phenology

## Abstract

The Indochinese Peninsula is a main center of speciation of *Capparis*, but the taxonomic treatment of the genus is still critical in this area. With regard to Vietnam, a discordant number of species was recorded by different authors during the last century, whereas various new species have been recently described. An updated checklist of the intrageneric taxa occurring in the country is here presented, including a new species from the island of Hon Tre, Khanh Hoa Province. The genus comprises in Vietnam 37 species, 9 subspecies and 3 varieties, all belonging to *Capparis* sect. *Monostichocalyx*. The study area, with 10 endemic species, is confirmed as one of the hotspots of the genus. Three lectotypes are also selected. The new species here described and illustrated, *C. oxycarpa*, is related to *C. pranensis,* differing in the few-flowered subumbels, narrower sepals, smaller petals, longer filaments and smaller, apiculate fruit; its affinities with related taxa and conservation status are discussed, and data on its ecology and phenology are given.

## 1. Introduction

The genus *Capparis* L. includes about 150 species [1], occurring in a wide range of habitats in the tropical and subtropical regions of the Old World, with outliers in the Mediterranean region and central Asia [2]. Jacobs [3] regarded the Indochinese Peninsula as a main center of speciation of the genus, reporting in this area 31 species, 7 of which are endemic. Recently, various new species have been described from Vietnam and Laos [4,5,6,7,8,9,10,11,12,13,14,15], as well as from other countries of southern Asia and the western Pacific, such as India [16,17], Thailand [18], China [19], Malaysia [20], Indonesia [21,22] and New Caledonia [23,24]. With regard to Vietnam, the taxonomic treatment of the genus is still critical, with a discordant number of intrageneric taxa recorded by different authors during the past century [3,25,26,27]. All the species reported from this country belong to *Capparis* sect. *Monostichocalyx* Radlk., while other sections of the genus, i.e., sect. *Capparis* L., sect. *Sodada* (Forssk.) Endl. and sect. *Busbeckea* (Endl.) Benth. & Hook.f., are not represented in the Indochinese area [3,28,29]. In particular, sect. *Capparis* is distributed from southern Europe, southwestern Asia and Africa eastwards to Australia and the Pacific; sect. *Sodada* in Africa and southwestern Asia; and sect. *Busbeckea* in Australia with outliers in India, Sri Lanka, Indonesia, New Guinea and the western Pacific.

The present paper is aimed at providing an updated checklist of the genus in Vietnam, based on herbarium investigations as well as on the available bibliographic sources. Furthermore, during floristic research carried out in southern Vietnam, an unidentified population of *Capparis* was observed on Hon Tre Island, Khanh Hoa Province. Based on herbarium investigations, this population turned out to belong to a new species of sect. *Monostichocalyx*. The new species is here described, and its conservation status and affinities are discussed.

## 2. Results

In the present treatment, 37 species, besides 9 subspecies and 3 varieties, are recognized in Vietnam. Among these, a new species, *Capparis oxycarpa*, is here described and illustrated. As above mentioned, all the species occurring in the country belong to *Capparis* sect. *Monostichocalyx*, which comprises several species widespread from tropical Africa to the Pacific [3]. In the following checklist for each taxon are reported the types, synonyms, distribution and the codes of the herbaria where representative specimens were examined. 


**Checklist of intrageneric taxa of *Capparis* in Vietnam**
1. ***Capparis acutifolia*** Sweet, Hort. Brit., ed. 2: 585 (1830).1.1. ***Capparis acutifolia*** subsp. ***acutifolia*** Sweet. Type: [icon] “C. acuminata (non Willd.) Lindl., Bot. Reg. 16 (1830) t. 1320” (lectotype selected by Jacobs [3], p. 427).*Capparis acuminata* Lindl., Edwards’s Bot. Reg. 16: t. 1320 (1830).*Capparis membranacea* Gardner & Champ. Hooker’s J. Bot. Kew Gard. Misc. 1: 241 (1849).*Capparis kikuchii* Hayata, Icon. Pl. Formosan. 3: 21 (1913).*Capparis leptophylla* Hayata, Icon. Pl. Formosan. 3: 22 (1913).*Capparis tenuifolia* Hayata, Icon. Pl. Formosan. 3: 23 (1913).Distribution: China, Taiwan, Laos; central Vietnam [3,9]. Recorded also in India, Bhutan and Thailand [19]. In Herb. LE, P.1.2. ***Capparis acutifolia*** subsp. ***sabiifolia*** (Hook.f. & Thomson) M. Jacobs, Blumea 12(3): 432 (1965). *Capparis sabiifolia* Hook.f. & Thomson, Fl. Brit. India 1(1): 179 (1872). Type: [INDIA] Khasia, 13 July 1850, Hook. f. & Thomson 1692 (lectotype K 000247295 digital image!, selected by Maurya et al. [30], p. 129; isolectotype K 000247296 digital image!).*Capparis vientianensis* Gagnep., Bull. Soc. Bot. France 85: 599 (1938).Distribution: India, China, Taiwan, Myanmar, Thailand, Laos; northern Vietnam. In Herb. LE, P.1.3. ***Capparis acutifolia*** subsp. ***viminea*** M. Jacobs, Blumea 12(3): 429 (1965). Type: Sikkim, Hooker f. 48 (lectotype K 000247292 digital image!, selected by Fici [10], in publ.; isolectotypes: GH 00042264 digital image!, K 000247291 digital image!, L 0035305 digital image!).*Capparis viminea* Hook.f. & Thomson, Fl. Brit. India 1(1): 179 (1872).*Capparis membranifolia* Kurz, J. Asiat. Soc. Bengal, Pt. 2, Nat. Hist. 43(2): 70 (1874).*Ficus marchandii* H.Lév., Repert. Spec. Nov. Regni Veg. 12: 533 (1913).Distribution: India, Sikkim, Bhutan, China, Myanmar, Thailand, Laos; northern, central and southern Vietnam. Recorded also from Cambodia [19]. In Herb. P.1.4. ***Capparis acutifolia*** subsp. ***obovata*** M. Jacobs, Blumea 12(3): 433 (1965). Type: [Vietnam] Tonkin, Bord du Grand Lac, Hanoi, Apr. 1935, *Pételot* 5686 (holotype A 00042263 digital image!; isotypes L 0035304 digital image!, P 04022331!, P04022332!, US 00100530 digital image!).*Capparis sunbisiniana* M.L.Zhang & G.C.Tucker, Fl. China 7: 441 (2008).Distribution: China, Myanmar, Thailand; northern and central Vietnam. In Herb. LE, P.2. ***Capparis annamensis*** (Baker f.) M. Jacobs, Blumea 12(3): 433 (1965). *Capparis grandiflora* var. *annamensis* Baker f., J. Nat. Hist. Soc. Siam 4: 127 (1921). Type: [Vietnam] Annam, Tourcham, May 1918, *Kloss* s.n. (holotype BM, not seen).Distribution: Central Vietnam. In Herb. LE, P.3. ***Capparis assamica*** Hook. f. & Thomson, Fl. Brit. India 1: 177 (1872). Type: [India] East Bengal, *Griffith* 602 (lectotype K 000247299 digital image!, selected by Jacobs [3], p. 434).*Capparis gallatlyi* King, Ann. Roy. Bot. Gard. (Calcutta) 5: 118.Distribution: India, Bhutan, China, Myanmar, Thailand, Laos [19,31]; northern Vietnam [27].4. ***Capparis bachii*** Sy, R.K.Choudhary & Joongku Lee, Ann. Bot. Fenn. 55: 31 (2017). Type: Vietnam, Dong Nai Province, Tan Phu District, Ta Lai Community, 8 March 2013, *S.D. Thuong* 74 (holotype HN!; isotype HNU!).Distribution: Southern Vietnam. In Herb. HN, HNU.5. ***Capparis beneolens*** Gagnep., Bull. Soc. Bot. France 85: 597 (1938). Type: [Vietnam] Annam, Ca-Na, 24 Dec. 1923, *Poilane* 9320 (lectotype P 04746233!, selected by Jacobs [3], p. 436, as the holotype; isolectotypes A 00042266 digital image!, K 000651012 digital image!, L 0035306 digital image!).Distribution: Central Vietnam. In Herb. A, K, L, P.6. ***Capparis cantoniensis*** Lour., Fl. Cochinch. 1: 330 (1790). Type: [China] Kwang Tung Province, Canton & Vicinity, 9 March 1917, *Levine* 1247 (neotype E 00327222 digital image!, selected by Jacobs [3], p. 441; isoneotypes A 00277170 digital image!, GH 00042251 digital image!, K 000380489 digital image!, L 0035314 digital image!, US 00100536 digital image!).*Olofuton racemosum* Raf., Sylva Tellur. 108 (1838).*Capparis salaccensis* Blume, Bijdr. Fl. Ned. Ind. 2: 54 (1825).*Capparis pumila* Champ. ex Benth., Hooker’s J. Bot. Kew Gard. Misc. 3: 260 (1851).*Capparis sciaphila* Hance, Ann. Sci. Nat., Bot. sér. 5, 5: 206 (1866).*Capparis hasseltiana* Miq., Illustr. Fl. Arch. Ind. 24, t. 13 (1870).*Capparis celebica* Miq., Illustr. Fl. Arch. Ind. 26 (1870).*Capparis ambigua* Kurz, Forest Fl. Burma 1: 65 (1877).*Cudrania bodinieri* H.Lév., Repert. Spec. Nov. Regni Veg. 13: 265 (1914).Distribution: India (including Andaman Islands), Bhutan, Sikkim, China, Indonesia, Myanmar, Philippines, Thailand, Laos; northern, central and southern Vietnam. In Herb. P.7. ***Capparis daknongensis*** Sy, G.C.Tucker, Cornejo & Joongku Lee, Ann. Bot. Fenn. 50: 99 (2013). Type: Vietnam, Dak Nong Province, Gia Nghia District, Dak Nia Commune, 5 Apr. 2010, *TT. Bach, V.T. Chinh, D.V. Hai, B.H. Quang, S.D. Thuong* VK-3675 (holotype HN!; isotype KRIB!).Distribution: Southern Vietnam. In Herb. HN.8. ***Capparis dongvanensis*** Sy, B.H.Quang & D.V.Hai, Nordic J. Bot. 35: 272 (2016). Type: Vietnam, Ha Giang Province, Dong Van District, 23 March 2015, *T. T. Bach* et al. *Thuong* 73 (holotype HN!; isotype KRIB!).Distribution: Northern Vietnam. In Herb. HN.9. ***Capparis diffusa*** Ridl., J. Straits Branch Roy. Asiat. Soc. 59: 68 (1911). Type: [Malaysia] Perlis, Bukit Lagi, March 1910, *Ridley* 15174 (lectotype SING 0056837 digital image!, selected by Jacobs [3], p. 447, as the holotype; isolectotype K 000643988 digital image!).Distribution: Malaysia, Thailand, Indonesia; southern Vietnam. In Herb. P.10. ***Capparis erycibe*** Hall., Bull. Herb. Boiss. 6: 216 (1896). Type: [Indonesia] Java, Tjampea, Mt Tjibodas, Buitenzorg, 4 May 1895, *Hallier f.* 779^a^ (lectotype BO, selected by Jacobs [3], p. 449, as the holotype, not seen; isolectotype L 0035324 digital image!). In Herb. LE, P.*Capparis paniculata* Ridl., J. Fed. Malay States Mus. 10: 129 (1920).Distribution: Malaysia, Thailand, Indonesia; central Vietnam. Recently recorded also from Laos [32].11. ***Capparis fengii*** B.S. Sun, Acta Phytotax. Sin. 9(2): 113 (1964). Type: [China] Yunnan, Wen-Shan Hsien, 5 May 1962, *K.M. Feng* 22452 (holotype KUN 0494378 digital image!; isotype IBSC 0134488 digital image!).Distribution: China; northern Vietnam [33]. In Herb. HN.12. ***Capparis flavicans*** Kurz, J. Asiat. Soc. Bengal, Pt. 2, Nat. Hist. 39(2): 62 (1870). Type: [Thailand] Campong Kankian by Radboerie [Ratchaburi], *Teijsmann* 5931 (lectotype CAL, selected by Jacobs [3], p. 451, as the holotype, not seen; isolectotypes K 000651014 digital image!, U 1164806 digital image!).*Capparis flavicans* Wall. ex Hook.f. & Thomson, Fl. Brit. India 1(1): 179 (1872), nom. illeg.*Capparis cambodiana* Pierre ex Gagnep., Bull. Soc. Bot. France 55: 210 (1908).Distribution: Myanmar, Thailand, Laos, Cambodia; central and southern Vietnam. Doubtfully recorded from a single locality in southern India by Jacobs [3]. In Herb. P.13. ***Capparis floribunda*** Wight, Ill. Ind. Bot. pts. 1-8: 35, t. 14 (1838). Type: [India] Peninsula Ind. orientalis, *Wight* 2439 (lectotype K 000247308 digital image!, selected here; isolectotypes E 00179031 digital image!, K 000247309 digital image!, NY 00387644 digital image!).*Crateva octandra* Blanco, Fl. Filip. [F.M. Blanco] 400 (1837).*Capparis oligandra* Griff., Not. Pl. Asiat. 4: 577 (1854).*Capparis luzonensis* Turcz., Bull. Soc. Imp. Naturalistes Moscou 27(2): 324 (1854).*Capparis andamanica* King, Ann. Roy. Bot. Gard. (Calcutta) 5: 119.*Capparis oligostema* Hayata, Icon. Pl. Formosan. 3: 22 (1913).Distribution: India (including Andaman Islands), Sri Lanka, Myanmar, Thailand, Taiwan, Philippines, Indonesia; central Vietnam [3]. In Herb. P.Note: *C. floribunda* Wight was described from material collected by the same author in India. Treating this species Jacobs [3] wrote “Type: *Wight propr.* 2439 (K)”, thereby designating the Wight’s material in K as the lectotype (first-step). Since two specimens of this gathering are present in K (barcodes K 000247308 and K 000247309), the specimen K 000247308 is here designated as the (second-step) lectotype (Art. 9.17 of *International Code of Nomenclature for algae, fungi, and plants* [34]).14. ***Capparis gialaiensis*** Sy, Ann. Bot. Fenn. 52: 219 (2015). Type: Vietnam, Gia Lai Province, K’Bang District, Son Lang Commune, 23 May 2013, *T.T. Bach* et al. VK 5402 (holotype HN!; isotype KRIB!).Distribution: Central Vietnam. In Herb. HN.15. ***Capparis grandis*** L.f., Suppl. Pl. 263 (1782). Type: [India], s. loc., *König* s.n. (lectotype C 10009043 digital image!, selected by Maurya et al. [30], p. 126).*Capparis maxima* Roth, Sp. Pl. Nov. 237 (1821).*Capparis obovata* Buch.-Ham. ex DC., Prodr. 1: 248 (1824).*Capparis racemifera* DC., Prodr. 1: 248 (1824).*Capparis bisperma* Roxb., Fl. Ind. 2: 568 (1832).Distribution: India, Sri Lanka, Myanmar, Thailand; central and southern Vietnam [3]. In Herb. P.16. ***Capparis kbangensis*** Sy & D.V.Hai, PhytoKeys 151: 84 (2020). Type: Vietnam, Gia Lai Province, Kbang District, 7 April 2018, *Sy Danh Thuong, Do Van Hai, Thuong* 0704201801 (holotype HN!; isotype IBSC!).Distribution: Central Vietnam. In Herb. HN.17. ***Capparis khuamak*** Gagnep., Bull. Soc. Bot. France 85: 598 (1938). Type: Laos, Sam-Neua, 8 Oct. 1920, Poilane 2005 (lectotype P 04022035!, selected by Jacobs [3], p. 457, as the holotype; isolectotype L 0035330 digital image!).*Capparis trichopoda* B.S.Sun, Acta Phytotax. Sin. 9(2): 116 (1964).Distribution: China, Laos; central and southern Vietnam. In Herb. L, P.18. ***Capparis koioides*** M. Jacobs, Blumea 12(3): 459 (1965). Type: [Vietnam] Sud-Annam, Province de Nhatrang, Massif du Hon Bà, 25 Aug. 1918, *Chevalier* 38761 (lectotype P 05453686!, selected here; isolectotype P 05453685!).Distribution: Thailand; central Vietnam. In Herb. P.Note: *C. koioides* was described by Jacobs [3], who indicated the type as “*Chevalier* 38761 (L, phot., P, holo)”. In the herbarium P there are two duplicates of this gathering (barcodes P 05453685 and P 05453686). The name is validly published because a single gathering was indicated by Jacobs as type (Art. 40.2 of *International Code of Nomenclature for algae, fungi, and plants* [34]); the two specimens in P are syntypes (Art. 9.6 and Art. 40.2, Note 1 [34]), and one of them (P 05453686) is here designated as the lectotype.19. ***Capparis longestipitata*** Heine, Mitt. Bot. Staatssamml. München 1(6): 210 (1953). Type: [Malaysia], North Borneo, Mt. Kinabalu, Tenompok 1500 m, 7 June 1932 *Clemens* 29812 (lectotype M, selected by Jacobs [3], p. 465, as the holotype, not seen; isolectotypes A 00042279 digital image!, BM 000541238 digital image!, L 0035337 digital image!, NY 00387658 digital image!).Distribution. Indonesia; central Vietnam. In Herb. P.20. ***Capparis micracantha*** DC., Prodr. 1: 247 (1824). Type: [Indonesia] Java, *Lahaye* s.n. (lectotype G 00203273 digital image!, selected by Jacobs [3], p. 467, as the holotype).20.1 ***Capparis micracantha*** subsp. ***micracantha*** var. ***micracantha****Capparis callosa* Blume, Bijdr. Fl. Ned. Ind. 2: 53 (1825).*Capparis odorata* Blanco, Fl. Filip. 439 (1837).*Capparis forsteniana* Miq., Illustr. Fl. Arch. Ind. 32 (1870).*Capparis hainanensis* Oliv., Hooker’s Icon. Pl. 16: t. 1588 (1887).*Capparis myrioneura* Hallier f., Repert. Spec. Nov. Regni Veg. 2: 60 (1906).*Capparis bariensis* Pierre ex Gagnep., Bull. Soc. Bot. France 55: 209 (1908).*Capparis donnaiensis* Pierre ex Gagnep., Bull. Soc. Bot. France 55: 211 (1908).*Capparis venosa* Merr., Philipp. J. Sci., C 10: 305 (1915).*Capparis liangii* Merr. & Chun, Sunyatsenia 2: 29 (1934).*Capparis petelotii* Merr., J. Arnold Arbor. 23: 166 (1942).Distribution: India (Andaman Islands), Myanmar, China, Laos, Cambodia, Thailand, Malaysia, Indonesia, Philippines; northern, central and southern Vietnam [3]. In Herb. K, L, LE, P.20.2 ***Capparis micracantha*** subsp. ***micracantha*** var. ***henryi*** (Matsum.) Jacobs, Blumea 12(3): 470 (1965). *Capparis henryi* Matsum., Bot. Mag. (Tokyo) 13: 33 (1899). Type: Formosa [Taiwan], Takow, *A. Henry* 570 (lectotype TI, selected by Jacobs [3], p. 470, as the holotype, not seen; isolectotypes A 00042255 digital image!, A 00042256 digital image!, K 000380497 digital image!, NY 00387639 digital image!, US 00100552 digital image!, US 00100553 digital image!).Distribution: Taiwan [3]; southern Vietnam [26,27].20.3 ***Capparis micracantha*** subsp. ***korthalsiana*** (Miq.) M. Jacobs, Fl. Males., Ser. 1, Spermat. 6(1): 86 (1960). *Capparis korthalsiana* Miq., Ill. Fl. Archip. Ind.: 31 (1870). Type: [Indonesia] Borneo, Pulu Lampei, *Korthals* s. n. (lectotype L 0035343 digital image!, selected by Jacobs [3], p. 471).*Capparis finlaysoniana* Wall. ex Hook.f. & Thomson, Fl. Brit. India 1(1): 179 (1872).Distribution: Malaysia, Singapore, Indonesia [3]; southern Vietnam [26,27].21. ***Capparis oxycarpa*** Fici, Averyanov & Sy, **sp. nov**. Type: Vietnam, Phu Khanh [Khanh Hoa Province], Nha Trang, Hon Tre Island, 12°11’ N, 109°16’ E, 6 xi 1989, *Averyanov & Kudryavtzeva* 83 (holotypus LE 01077298!; isotypus LE 01077300!).Distribution: Southern Vietnam. In Herb. LE.22. ***Capparis pranensis*** (Gagnep.) M. Jacobs, Blumea 12(3): 477 (1965). *Capparis thorelii* var. *pranensis* Gagnep., Bull. Soc. Bot. France 55(3): 214 (1908). Type: [Thailand] ad Muong Prang in peninsula Malayana, Aug. 1868, *Pierre* 4018 (lectotype P 04022393!, selected by Fici [10], in publ.; isolectotypes P 04022391, P 04022392 digital images!, K 000651018 digital image!, LE 00013167 digital image! MPU 600749 digital image!).Distribution: Thailand, Cambodia; central Vietnam [3]. In Herb. L, P.23. ***Capparis pubiflora*** DC., Prodr. 1: 246 (1824). Type: [Indonesia] Timor, *Anonymous* s.n. (lectotype P, selected by Jacobs [3], p. 479, as the holotype, not seen; isolectotype G 00207275 digital image!).*Capparis nigricans* Span., Linnaea 15(2): 165 (1841).*Capparis cerasifolia* A.Gray, U.S. Expl. Exped., Phan. 15: 71 (1854).*Capparis brachyscias* Turcz., Bull. Soc. Imp. Naturalistes Moscou 27(2): 323 (1854).*Capparis lasiopoda* Turcz., Bull. Soc. Imp. Naturalistes Moscou 27(2): 322 (1854).*Capparis myrioneura* Hallier f., Repert. Spec. Nov. Regni Veg. 2: 60 (1906).*Capparis borneensis* Merr., Univ. Calif. Publ. Bot. 15: 91 (1929).*Capparis braianensis* Gagnep., Bull. Soc. Bot. France 85: 597 (1939).Distribution: China, Thailand, Malaysia, Indonesia, Philippines; northern and central Vietnam [3]. In Herb. P.24. ***Capparis pubifolia*** B.S. Sun, Fl. Yunnan. 2: 64 (1979). Type: [CHINA] *Chang* 13028 (holotype KUN 0137208 digital image!).Distribution: China; recently recorded in northern Vietnam [35]. In Herb. HN, VFM.25. ***Capparis pyrifolia*** Lam., Encycl. 1(2): 606 (1785). Type: India, *Poivre* s.n. (lectotype P 00680459!, selected by Jacobs [3], p. 480).*Capparis acuminata* Willd., Sp. Pl., ed. 4, 2(2): 1131 (1799).*Capparis foetida* Blume, Bijdr. Fl. Ned. Ind. 2: 52 (1825).*Capparis dasypetala* Turcz., Bull. Soc. Imp. Naturalistes Moscou 27(2): 322 (1854).*Capparis oxyphylla* Miq., Pl. Jungh. 4: 397 (1855).*Capparis kerrii* Craib, Bull. Misc. Inform. Kew 1922(8): 232 (1922).Distribution: Thailand, Laos, Cambodia, Indonesia; central and southern Vietnam. In Herb. K, P.26. ***Capparis radula*** Gagnep., Bull. Soc. Bot. France 55: 213 (1908). Type: [Laos], Bassac [Champasak], Jan. 1877, Harmand 1094 (lectotype P 04022374!, selected by Fici [10], in publ.; isolectotype P 04022375!).Distribution: Thailand, Laos, Cambodia; southern Vietnam. In Herb. P.27. ***Capparis rigida*** M. Jacobs, Blumea 12(3): 485 (1965). Type: [Vietnam] Annam, Ca-Na, Prov. Phanrang, 10 Dec. 1923 *Poilane* 9081 (holotype P 04022373!).Distribution: Central Vietnam. In Herb. L, P.28. ***Capparis sepiaria*** L., Syst. Nat., ed. 10. 2: 1071 (1759). Type: “Ind. hab. ad sepes”, *Anonymous* s.n. (lectotype LINN-HL 664.4 digital image!, selected by Jacobs [3], p. 489).*Capparis incanescens* DC., Prodr. 1: 247 (1824).*Capparis umbellata* R.Br. ex DC., Prodr. 1: 247 (1824).*Capparis emarginata* C.Presl, Reliq. Haenk. 2: 85 (1835).*Capparis retusella* Thwaites, Enum. Pl. Zeyl. 16 (1858).*Capparis glauca* Wall. ex Hook.f. & Thomson, Fl. Brit. India 1(1): 180 (1872).*Capparis flexicaulis* Hance, J. Bot. 16: 225 (1878).*Capparis trichopetala* Valeton, Bull. Dépt. Agric. Indes Néerl. 10: 14 (1907).*Capparis affinis* Merr., Philipp. J. Sci., C 10: 303 (1915).Distribution: Africa: Mauritania, Senegal, Mali, Côte d’Ivoire, Burkina Faso, Niger, Nigeria, Benin, Cameroon, Chad, Central African Republic, Sudan, Ethiopia, Eritrea, Somalia, Kenya, Tanzania, Uganda, Rwanda, Burundi, Democratic Republic of Congo, Malawi, Zambia, Angola, Botswana, Zimbabwe, Mozambique, South Africa, Madagascar; Australia: Northern Territory, Western Australia, Queensland; Asia: India (including Andaman Islands), Sri Lanka, Nepal, Myanmar, China, Laos, Cambodia, Thailand, Malaysia, Indonesia, Philippines, New Guinea; northern, central and southern Vietnam [3,10,19]. In Herb. L, LE, P.29. ***Capparis siamensis*** Kurz, Forest Fl. Burma 1: 63 (1877). (1877: 63). Type: [Thailand] Siam, Radboore [Ratchaburi], *Teijsmann* 5927 (lectotype CAL, selected by Jacobs [3], p. 493, as the holotype, not seen; isolectotypes GH 00042274 digital image!, K 000651017 digital image!, U 0000961 digital image!).*Capparis macropoda* Pierre ex Gagnep., Fl. Indo-Chine 1: 196 (1908).*Capparis adunca* Craib, Bull. Misc. Inform. Kew 1922(8): 231 (1922).*Capparis winitii* Craib, Bull. Misc. Inform. Kew 1922(8): 234 (1922).Distribution: Thailand, Cambodia; southern Vietnam [26,27].30. ***Capparis sikkimensis*** Kurz, J. Asiat. Soc. Bengal, Pt. 2, Nat. Hist. 43(3): 181 (1874).30.1 ***Capparis sikkimensis*** subsp. ***formosana*** (Hemsl.) M. Jacobs, Blumea 12(3): 497 (1965). *Capparis formosana* Hemsl., Ann. Bot., Lond. 9: 145 (1895). Type: [Taiwan] Bankinsing Mts and Ape’s Hill, 1894, *A. Henry* 501^A^ (holotype K 00038502 digital image!).Distribution: China, Taiwan, Japan; northern Vietnam [19].30.2 ***Capparis sikkimensis*** subsp. ***masakai*** (H.Lév.) M. Jacobs, Blumea 12(3): 496 (1965). *Capparis masakai* H.Lév., Fl. Kouy-Tcheou 59 (1914). Type: [China] Kouy Tchéou [Kweichow], Lo Kouen chemin de Pin Fa, 15 May 1912, *Esquirol* 3230 (lectotype E 00327218 digital image!, selected by Jacobs [3], p. 496; isotype A 00251638 digital image!).Distribution: China; northern Vietnam [36]. In Herb. HN.30.3 ***Capparis sikkimensis*** subsp. ***yunnanensis*** (Craib & W.W.Sm.) M. Jacobs, Blumea 12(3): 496 (1965). *Capparis yunnanensis* Craib & W.W.Sm., Notes Roy. Bot. Gard. Edinburgh 9: 91 (1916). Type: [China] Yunnan, Szemao, *A. Henry* 12986 (lectotype E 00217925 digital image!, selected by Jacobs [3], p. 496, as the holotype; isolectotypes A 00042262 digital image!, K 000380499 digital image!, K 000380500 digital image!, US 00100597 digital image!).*Capparis bhamoensis* Raizada, Indian Forest Rec., Bot. 3: 127 (1941).Distribution: China, Myanmar, Thailand; northern Vietnam [3,19]. In Herb. P.31. ***Capparis subsessilis*** B. S. Sun, Acta Phytot. Sin. 9: 110 (1964). Type: [China] Kwangsi, Lung-Tsin Hsien, 22 Oct. 1956, *T. D. Li* 3112 (holotype IBK 00406871 digital image!; isotype GAC 0002093 digital image!).Distribution: China; northern Vietnam [19]. In Herb. LE.32. ***Capparis thorelii*** Gagnep., Bull. Soc. Bot. France 55: 214 (1908). Type: [Cambodia] Oudan [Oudong], 1866–1868, Thorel 2037 (lectotype P 04022397!, selected by Fici [10], in publ.; isolectotypes P 04022396!, P 04022398!, K 000651019!, L 0035361 digital image!, MPU 600750 digital image!).Distribution: Thailand, Cambodia; central Vietnam. In Herb. L, P.33. ***Capparis tonkinensis*** Gagnep., Bull. Soc. Bot. France 55: 215 (1908). Type: [Vietnam] Tonkin, in montibus Làng Hê, 18 Oct. 1888, *Bon* 4016 (lectotype P 04022385!, selected by Jacobs [3], p. 499, as the holotype; isotype L 0035362 digital image!).*Capparis indochinensis* Merr., Univ. Calif. Publ. Bot. 10: 424 (1924).Distribution: Northern and central Vietnam. In Herb. L, P.34. ***Capparis trinervia*** Hook.f. & Thomson var. ***trinervia***, Fl. Brit. India 1(1): 175 (1872). Type: [Myanmar] Tenasserim, *Helfer* 185 (lectotype K 000247349 digital image!, selected by Jacobs [3], p. 500).*Capparis kunstleri* King, J. Asiat. Soc. Bengal, Pt. 2, Nat. Hist. 58(4): 396 (1889).Distribution: Myanmar, Malaysia, Laos, Indonesia; northern and central Vietnam. In Herb. L, P.35. ***Capparis versicolor*** Griff., Not. Pl. Asiat. 4: 577 (1854). Type: [Myanmar] Mergui, Jan. 1835, *Griffith* 936 (lectotype K 000247351 digital image!, selected here; isolectotype K 000247350 digital image!).*Capparis larutensis* King, J. Asiat. Soc. Bengal, Pt. 2, Nat. Hist. 58(4): 393 (1889).*Capparis koi* Merr. & Chun, Sunyatsenia 2: 28 (1934).*Capparis nhatrangensis* Gagnep., Bull Soc. Bot. France 85: 598 (1939).Distribution: China, Myanmar, Malaysia; central Vietnam [3]. Recorded also in India (Assam) and Thailand [19]. In Herb. LE, P.Note: *C. versicolor* was described by Griffith from material collected by the same author in Mergui (Myanmar). Treating this species Jacobs [3] wrote “Type: *Griffith* 936 (K)”, designating the Griffith material in K as the lectotype (first-step). Since two specimens of this gathering are present in K (barcodes K 000247350 and K 000247351), the specimen K 000247351 is here designated as the (second-step) lectotype (Art. 9.17 of *International Code of Nomenclature for algae, fungi, and plants* [34]).36. ***Capparis viburnifolia*** Gagnep., Bull. Soc. Bot. France 85: 598 (1938). Type: [Vietnam] Tonkin, Seoleng piste de Phông Thô à Song tong Ngai, prov. de Lai Chau, 7 Apr. 1936, *Poilane* 25565 (holotype P 04022395!).Distribution: China, Thailand; northern and central Vietnam. In Herb. L, P.37. ***Capparis zeylanica*** L., Sp. Pl., ed. 2, 1: 720 (1762). Type: Ceylon [Sri Lanka], *Hermann* 210 (lectotype BM, selected by Jacobs [3], p. 505), not seen).*Capparis horrida* L.f., Suppl. Pl. 264 (1782).*Capparis dealbata* DC., Prodr. 1: 246 (1824).*Capparis quadriflora* DC., Prodr.1: 247 (1824).*Capparis terniflora* DC., Prodr. 1: 247 (1824).*Capparis aurantioides* C.Presl, Reliq. Haenk. 2: 86 (1835).*Capparis linearis* Blanco, Fl. Filip. 438 (1837).*Capparis nemorosa* Blanco, Fl. Filip. 438 (1837).*Capparis rufescens* Turcz., Bull. Soc. Imp. Naturalistes Moscou 27(2): 321 (1854).*Capparis erythrodasys* Miq., Pl. Jungh. 4: 397 (1855).*Capparis hastigera* Hance, J. Bot. 6: 296 (1868).*Capparis swinhoii* Hance, J. Bot. 6: 296 (1868).*Capparis crassifolia* Kurz, J. Asiat. Soc. Bengal, Pt. 2, Nat. Hist. 42(4): 227 (1874).*Capparis polymorpha* Kurz, J. Asiat. Soc. Bengal, Pt. 2, Nat. Hist. 42(4): 227 (1874).*Capparis xanthophylla* Coll. & Hemsl., J. Linn. Soc., Bot. 28: 20 (1890).*Capparis latifolia* Craib, Bull. Misc. Inform. Kew 1922(8): 232 (1922).*Capparis subhorrida* Craib, Bull. Misc. Inform. Kew 1922(8): 234 (1922).Distribution: India (including Andaman Islands), Sri Lanka, Nepal, China, Myanmar, Thailand, Laos, Cambodia, Philippines, Indonesia; northern, central and southern Vietnam. In Herb. L, P.

### ***Morphological Analysis of the Hon Tre* Capparis *Population*** 

Table 1 describes the main features of the *Capparis* population from Hon Tre Island and compares them with those of the related taxa belonging to the Subumbellates-Group [3], recorded from Vietnam. As reported in the diagnosis furnished in the Taxonomic treatment, the studied population is mainly related to *C. pranensis*, a species known from Thailand, Cambodia and Vietnam [3,28], resembling it in the vegetative features but differing in various characters of the inflorescence, flower and fruit (Table 1). The Hon Tre population also shows affinities with *C. thorelii* Gagnep., recorded from Thailand, Cambodia and Vietnam, which is distinguished by the twigs pubescent, later glabrescent, subumbels up to 8-flowered often arranged in a panicle, stamens c. 35 with filaments c. (6–) 7–10 mm long and fruit 1.1–1.8 cm in diameter, not apiculate. In addition, *C. sepiaria* L., a wide-ranging species distributed from tropical Africa eastwards to southern Asia and Australia [3], shows some affinities with the studied population, differing in the twigs pubescent, later glabrescent, leaves mostly pubescent in the lower surface, subumbels up to 26-flowered, filaments 0.5–1 cm long, and fruit not apiculate. Several other species of the Subumbellates-Group, reported from Vietnam, are easily distinguished by both vegetative and reproductive characters (Table 1).

## 3. Discussion

As mentioned above, the diversity of the genus *Capparis* is still not exhaustively investigated in Vietnam, where a discordant number of species was recorded in the past by different authors. In particular, Gagnepain [25] reported 13 species from the area, while more recently Ho [26] and Ban and Dorofeev [27] respectively reported 34 and 30 species. Furthermore, during the past decade various new species were described and a few others were first recorded from Vietnam. Five new species, *C. bachii, C. daknongensis*, *C. dongvanensis*, *C. gialaiensis* and *C. kbangensis*, were described by Sy et al. [4,5,6,7,8] from different areas of the country, while three taxa, *C. fengii*, *C. pubifolia* and *C. sikkimensis* subsp. *masakai*, were first recorded [33,35,36]. Based on the checklist presented here, the genus includes in Vietnam 37 species, 9 subspecies and 3 varieties. With regard to the whole Asiatic continent, a comparable richness in species is reported from China with 37 species [19], and India with 34 species [37], while among the other countries of southeastern Asia, 26 species are recorded from Thailand [18,31], 22 from Laos [10], 11 from Malaysia [20] and 9 from Cambodia [10]. All the species of *Capparis* occurring in Vietnam belong to sect. *Monostichocalyx*. Within this section Jacobs [3], in his revision of the genus from the Indus to the Pacific, recognized some “tentative” groups, mainly distinguished by the inflorescence and flower characters; most species reported from Vietnam (20) belong to the Subumbellates-Group, which is characterized by flowers more or less neatly subumbellate, flowers medium- to small-sized (sepals 2–10 mm long), stamens under 70, ovary 1–3 mm long and 2–4 placentas [3]; in addition, 6 species belong to the Seriales-Group, 5 to the Brevispina-Group, 2 to the Trinervia-Group and Cataphyllosa-Group, and 1 to the Moonii-Group and Grandis-Group. Among the 37 species of *Capparis* occurring in Vietnam, 10 (27%) are endemics, i.e., *C. annamensis*, *C. bachii*, *C. beneolens*, *C. daknongensis*, *C. dongvanensis*, *C. gialaiensis*, *C. kbangensis*, *C. oxycarpa*, *C. rigida* and *C. tonkinensis*. Jacobs [3], p. 403, underlined that the endemicity is generally low within the genus, and regarded “the lowlands of the Indo-Chinese Peninsula, particularly of SE. Annam” as one of the main centers of speciation of *Capparis*; this statement is confirmed by our data, with seven species endemic to central Vietnam. Furthermore, the study of the historical herbarium collections carried out during the present research allowed us to designate the lectotypes of three species, i.e., *C. floribunda*, *C. koioides* and *C. versicolor*.

With regard to the new species here described, *C. oxycarpa*, it belongs to the Subumbellates-Group [3] and is easily distinguished from other taxa of the same group by the few-flowered inflorescence, with flowers in some cases solitary, and by the small, apiculate fruit. The latter character is so far known only from a few species of the genus, belonging to other groups of the ones recognized by Jacobs [3] within *Capparis.* As reported above, it is mainly related to *C. pranensis* in its vegetative characters, but differs from this species in various reproductive features (Table 1); furthermore, also the flowering period is differentiated, i.e., September in *C. oxycarpa* and March in *C. pranensis* [3]. The new species is so far known only from Hon Tre, an island lying a short distance from the Vietnamese coast, with highest elevation of about 460 m. Despite being influenced by anthropic impacts, large areas of Hon Tre Island are still covered by well-preserved forest communities. Finally, it is to be underlined that, based on herbarium collections, the genus *Capparis* is represented on this island by a few other species, i.e., *C. floribunda* Wight, *C. micracantha* DC. and *C. zeylanica* L.

## 4. Taxonomic Treatment

### **Capparis oxycarpa**
*Fici, Averyanov & Sy, **sp. nov.*** 

Diagnosis: Related to *Capparis pranensis* (Gagnep.) M. Jacobs, but differing in the subumbels 2–3-flowered, in some cases the flowers solitary (vs. subumbels up to 10-flowered, often arranged in a panicle), sepals 2–3.1 mm wide (vs. 4–5.5 mm), petals 5.2–6.2 × 1.9–2.1 mm (vs. 7–9 × 3–4 mm), filaments 1–1.6 cm long (vs. 0.7–0.9 cm) and fruit 4–10 mm in diameter, markedly apiculate (vs. 11–30 mm, not apiculate).

Type: Vietnam, Phu Khanh [Khanh Hoa Province], Nha Trang, Hon Tre Island, 12°11’ N, 109°16’ E, 6 xi 1989, *Averyanov & Kudryavtzeva* 83 (holotypus LE 01077298!; isotypus LE 01077300!).

Description (Figure 1): Climber 1.5–2 m tall. Twigs zig-zag, glabrous. Stipular thorns recurved, 1–5 mm long, vigorous. Petiole 2–5 mm long. Leaf blade elliptic or ovate, rarely obovate, 1.3–3.2 times as long as wide, widest at the middle or below, 2.3–5.7 × 1.4–3 cm; base rounded or subcordate, apex rounded or subemarginate, rarely obtuse; surfaces glabrous; nerves 4–7 pairs. Subumbels 2–3-flowered, at the top of short lateral twigs or terminal, in some cases the flowers solitary; bracts 0.7–0.8 mm long or lacking; pedicels 0.6–2.2 cm long; sepals 3.5–5.5 × 2–3.1 mm, the outer pair coriaceous, elliptic, concave or navicular, obtuse, glabrous, ciliate at the margins, the inner pair thinner, obovate, slightly concave, rounded at apex, glabrous, membranaceous and villose at the margins; petals narrowly obovate to spatulate, 5.2 –6.2 × 1.9–2.1 mm, densely villose inside, villose at the base outside; stamens c. 24–26, filaments 1–1.6 cm long; gynophore 0.9–1.4 cm long, glabrous; ovary ovoid or ellipsoid, 0.9–1.2 × 0.8–0.9 mm long, glabrous, shortly beaked. Fruit ovoid, 5.5–12 × 4–10 mm, sharply apiculate, pericarp smooth, on thickened stipe.

Etymology: The specific epithet is composed of the Greek words *oxýs* (ὀξύς), meaning sharp, and *karpós* (καρπóς), fruit.

Distribution and habitat: The new species is only known from Hon Tre Island (Figure 2), where it has been observed in two close locations, in xerophytic scrub on sea faced dry slopes and in secondary lowland forest, up to 100 m elevation.

Phenology: Based on the available material flowering occurs in September, fruiting in November.

Conservation status: The new species is so far known from two locations of Hon Tre, in an area of occupancy less than 2 km^2^, and its population is estimated to number fewer than 1000 mature individuals. Following the *IUCN Red List Categories and Criteria* [38], it is here assessed as Vulnerable (VU D1).

Additional specimens examined: Vietnam. Phu Khanh (Khanh Hoa Province), Nha Trang, Hon Tre Island, 6 xi 1989, *Averyanov & Kudryavtzeva* 178 (LE 01077299); Phu Khanh (Khanh Hoa Province), Hon Tre Island, 29 ix 1990, *Kudryavtzeva & Ogureeva* 134 (LE 01077297).

## 5. Materials and Methods

Herbarium investigations were carried out on the collections of *Capparis* kept at HN, HNU, LE and P, coupled with research on the available online collections at A, BM, E, G, GAC, GH, IBK, IBSC, K, KUN, L, LINN, MPU, NY, SING, U, US. The taxonomic treatment here adopted, as well as the main diagnostic characters within the genus, follow Jacobs [3]. The data on the distribution of the taxa are based on herbarium collections and on bibliographic sources [3,4,5,6,7,8,9,10,19,20,24,25,26,27,28,29,33,35,36]. The species recognized in the study area are arranged in alphabetical order in the checklist. The herbarium acronyms follow Thiers [39], while authors and names of plants are based on the *International Plant Names Index* [40]. The examination of type specimens was carried out through electronic images available at JSTOR Global Plants [41] and through study of the historical collections at P. The new species here described was first observed and collected by one of us (L.A.) in November 1989 and later during September 1990, in the frame of field investigations carried out in southern Vietnam; the collections were kept at the Komarov Botanical Institute (LE). The description and illustration are based on herbarium material. The terminology of vegetative and reproductive structures is based on Jacobs [3]. The measurements of morphometric characters were carried out on the herbarium material using a digital caliper and the range of variation of each character, from the minimum to the maximum value, was reported in the description and Table 1. The data concerning the taxa related to the new species, reported in Table 1, were obtained from bibliographic sources [3,10,42]. The conservation status of the new species has been provisionally assessed according to *IUCN Red List Categories and Criteria* [38].

## Figures and Tables

**Figure 1 plants-11-03402-f001:**
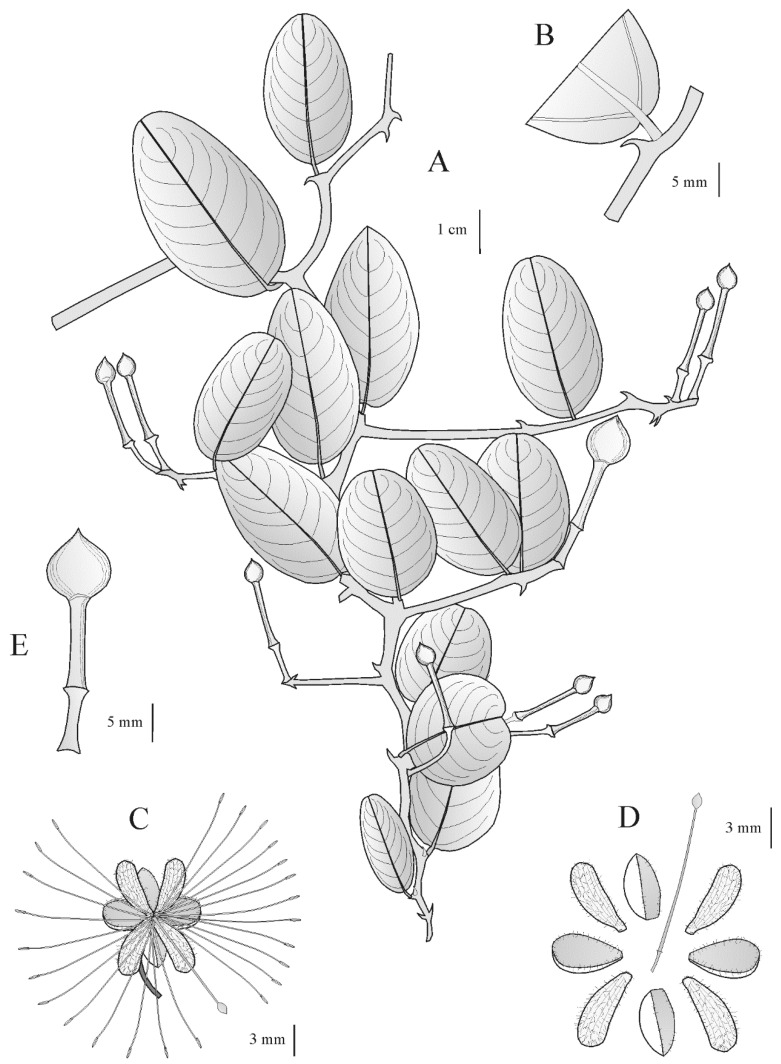
*Capparis oxycarpa*: (**A**) fruiting branch; (**B**) stipular thorn and base of leaf blade; (**C**) flower; (**D**) dissected flower showing sepals, petals, gynophore and ovary; (**E**) fruit and stipe; (**A**,**B**,**E**) from holotype (*Averyanov & Kudryavtzeva* 83); (**C**,**D**) from *Kudryavtzeva & Ogureeva* 134. Drawn by S. Fici.

**Figure 2 plants-11-03402-f002:**
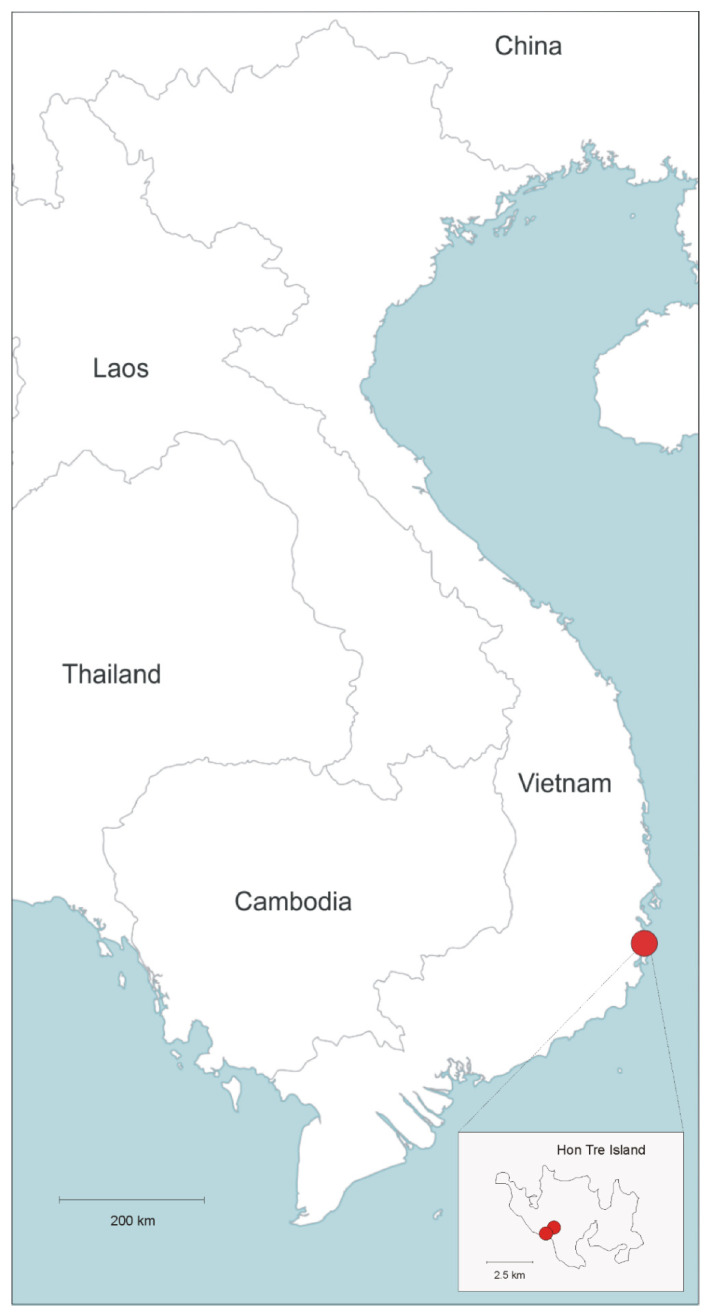
Known distribution of *Capparis oxycarpa* (red circles).

**Table 1 plants-11-03402-t001:** Features of the Hon Tre *Capparis* population compared with those of related species.

	Hon Tre Population	*C. pranensis*	*C. thorelii*	*C. sepiaria*	*C. diffusa*	*C. longestipiata*	*C. cantoniensis*	*C. tonkinensis*	*C. erycibe*	*C. floribunda*
Habit Height (m)	climber 1.5–2	shrub or climber up to 10	shrub or small tree, (or climber) up to 5	shrub (or climber) up to 3–4	shrub or climber up to c. 4	unknown	climber up to 20	climber up to 5	climber up to c. 14	shrub or climber up to c. 3–4
Indumentum of twigs	glabrous	glabrous	pubescent, later glabrescent	pubescent, later glabescent	pubescent, later glabrescent	pubescent, later glabrescent	puberulous, later glabrescent	puberulous, later glabrescent	puberulous	glabrous, rarely puberulous
Length of stipular thorns (mm)	1–5	2–5	3–5	(1–) 1.5–5 (–8)	1–3 rarely wanting	3–4 sometimes wanting	1–5 often wanting	1–3 sometimes wanting	up to 2 often wanting	1–2 often wanting
Length of petiole (mm)	2–5	4–7	3–6 (–7)	2–5 (–7)	2–4	c. 10	4–6 (–10)	3–8	4–10	5–15 (–20)
Leaf blade Size (cm) Length/width ratio Pubescence	2.3–5.7 × 1.4–3 1.3–3.2 glabrous	(2–) 3–6 (–7) × (1.8–) 2–3.5 (–4) (1.2–) 1.5–2.2 glabrous	(1.2–) 1.4–3 (–4) × (0.9–) 1–2.2 (–2.5) 1.2–2 (–2.5) glabrescent	(1.5–) 2–8.2 (–10) × (0.8–) 1.3–3.6 (–4) (1.4–) 1.7–2.7 (–4) lower surface mostly pubescent	4.2–7.5 (–8.5) × 1.5–4 (–5) c. 1.7–2.9 glabrous	5–7 × 2.5–3.5 c. 1.7–2.4 glabrous	(3–) 5.5–10.5 (–12.5) × (1.5–) 2–4 (–4.5) 2.3–4 (–5) lower surface sparsely puberulous	3.5–8.5 × 1.2–4 2–3.5 glabrous	(9.5–) 12–16 (–20) × 4.5–8.5 1.8–2.5 (–2.8) upper surface glabrous, lower glabrescent	(4–) 6–10 (–13) × (1.5–) 2.5–4 (–6) c. 2.5–3 glabrous
Leaf base	rounded or subcordate	rounded or obtuse	rounded or obtuse	rounded, subcordate acute or obtuse	rounded or obtuse	rounded	acute or obtuse	rounded, obtuse or subcordate	rounded, acute or obtuse	rounded, obtuse or acute
Leaf apex	rounded or subemargina- te, rarely obtuse	obtuse, rounded or emarginate, someti- mes mucronulate	rounded, sometimes subemarginate	rounded or acute, often subemarginate	obtuse or subacuminate, sometimes subemarginate	acuminate, with tip 0.5–1.5 cm lon	shortly acuminate, mucronate	acute or acuminate, mucronulate	rounded or acuminate, mucronate	obtuse or rounded with short mucro
Number of nerves on each side of the midrib	4–7	(3–) 4–6 (–8)	4–6	(4–) 5–7 (–9)	5–8	5–7	6–10 (–12)	5–7	c. 6–8	7–9 (–12)
Inflorescence	subumbels 2–3-flowered, at the top of lateral twigs or terminal, sometimes flowers solitary	subumbels up to 10-flowered, at the top of lateral twigs or terminal, often arranged in a panicle	subumbels up to 8-flowered, at the top of lateral twigs or terminal, often arranged in a panicle	subumbels up to 26-flowered, at the top of lateral twigs, sometimes terminal	umbels 3–5-flowe- red, terminal or at the top of lateral twigs, sometimes arranged in a panicle	subumbels up to c. 15-flowered, at the top of lateral twigs or terminal, arran- ged in a panicle	subumbels up to c. 8- flowered, axillary, often arranged in a panicle	subumbels up to c. 7-flowered, axilla- ry or terminal, or race- me up to 35-flowered, sometimes arranged in a panicle	terminal panicle	subumbels up to c. 15-flowered, at the top of lateral twigs or terminal, arran- ged in a panicle
Pedicel length (cm)	0.6–2.2	(0.7–) 0.8–1.4 (–1.6)	(0.6–) 0.8–1.3 (–1.7)	(0.7–) 1.3–2.4 (–3)	2–5	0.8–1.5	0.4–2	5–10	0.4–1.8	8–12
Sepals Size (mm) Pubescence	3.5 –5.5 × 2–3.1 glabrous, outer pair ciliate at margins, inner pair villose at margins	4.5–6 × 4–5.5 glabrous, inner pair sometimes ciliate at margins	4–5 (–5.5) × 2–3 glabrous, sometimes ciliate at margins	3–5 (–6) × 2–4 (–5) glabrous	3–5 × 3–4 glabrous, inner pair ciliate at margins	3–5.5 × 3–4 pubescent ouside	3–7 (–8) × 2–6 outer pair sometimes puberulous outside, inner pair ciliate at margins	2.5–4 × 2–3 glabrous or puberulous outside	4–6 × 2.5–3 outer pair sometimes puberulous	2–4 × 1.5–2.5 glabrous
Petals Size (mm) Pubescence	5.2–6.2 × 1.9–2.1 densely villose inside, villose outside at base	7–9 × 3–4 pubescent	4.5–6 (–7) × 2–3 pubescent inside at base	4.5–6 (–7.5) × 1.5–3 pubescent	c. 4.5–8 × 2 pubescent inside	4–6 × 2 glabrous or pubescent	3.5–6.5 × (1.5–) 2–3 (–4) pubescent	2.5–5 × 2–2.5 pubescent at margins	4.5–6 × 1–4 sometimes puberulous at base	3–5 × 1.5–2 glabrous
Number of stamens	c. 24–26	c. 29–33	c. 35	c. (20–) 25–45	12–15 (–20)	18–30	20–45	18–24	20–40	7–9 (–12)
Length of filaments (cm)	1–1.6	0.7–0.9	c. (0.6–) 0.7–1	0.5–1	c. 1.2	0.7–2	0.8–1.5 (–2.5)	c. 0.4–0.6	0.5–0.6 (–0.8)	0.6–0.8
Gynophore Length (cm) Pubescence	0.9–1.4 glabrous	(0.6–) 0.7–1.1 glabrous	(0.7–) 1–1.5 glabrous	(0.4–) 0.7–1.3 (–1.5) mostly puberulous at the base	0.9–1.8 glabrous	2–3 glabrous	0.4–1.2 glabrous	0.1–0.15 glabrous	0.2–0.5 glabrous	0.4–0.6 (–1) glabrous
Ovary Size (mm) Shape	0.9–1.2 × 0.8–0.9 ovoid or ellipsoid	c. 1.5 × 1 ovoid	1.1–1.5 × 0.6–1 ovoid	1.5–2 × 0.8–1 ovoid or ellipsoid	c. 1 × 1 subglobose	c. 1.5 × 1 subovoid	1–1.5 × 0.5–1 ellipsoid or pear- shaped	1.2–2 × 0.7–1 glabrous	2 × 1 ovoid or spindle-shaped	0.7–1.5 × 0.5–1 ovoid
Fruit Diameter (mm) Shape	4–10 ovoid, sharply apiculate	11–30 globose	11–18 globose	(3–) 4–10 (–12) globose or subglobose	c. 8 globose	unknown	10–15 globose or ellipsoid	6–10 subellipsoid	c. 15 globose	15–20 (–25) globose

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
