# Peer review of "An Updated Checklist of the Genus Capparis L. (Capparaceae) in Vietnam, including a New Species from Hon Tre Island"

_plants, 2022, doi:10.3390/plants11233402_

Round 1

Reviewer 1 Report

Introd.: The sections not in Indo-China, where do they occur? Add in manuscript.

Why are species 2-5  not in bold face?

I cannot judge whether the tables for comparison show the data clearly, how does the final printout look?

Author Response

Response to Reviewer 1

The suggested changes and additions were made:

R.: Introd.: The sections not in Indo-China, where do they occur? Add in manuscript.

A.: The distribution of the other sections was added in the manuscript (Introduction).

R.: Why are species 2-5  not in bold face?

A.: All the species are in bold face in the revised version of the manuscript.

R.: I cannot judge whether the tables for comparison show the data clearly, how does the final printout look?

A.: This problem is probably due to the large size of the table. However the final printout of the table looks fine.

Reviewer 2 Report

The work is very interesting and reports an updated checklist of species of the genus Capparis for Vietnam as well as a new species for this geographical region. The sensation is of a work that combines two separate objectives, on the one hand, a new species and on the other an updated checklist of the genus Capparis in Vietnam. These two components of the article remain largely separate except for the fact that the list reports the new species. Therefore it remains difficult to organize a homogeneous path and to move from one topic to another it is necessary to make leaps that create fractures in the logical path of the article. The best thing would be to separate the two objectives into two different articles to be able to devote greater attention and depth to the two topics, both worthy of attention. To maintain a single article, it is necessary to deal clearly with the two objectives in all sections, distinguishing Materials and Methods, Results, Discussion and Conclusions.

The new species

 In the Result section, there is the taxonomic treatment of the new species, which should be after the Discussion section because in the Result section you have to illustrate the systematics analysis, i.e. a comparison between the new population and other similar species in the genus, and after the Discussion section, where commenting the peculiarities of the new population (such as the Taxonomic remark included in the text), inserting the taxonomic treatment.

 The risk status was assigned without giving detailed information on the surface occupied, the number and dispersion of the population(s) and the number of individuals.

Some numbers are reported in the Results, particularly in the table. In Materials and Methodas, it is necessary to explain what they indicate (mean, maximum, minimum, percentile, etc.) and on what basis they were obtained, whether on a statistical basis (number of populations, number of individuals and number of measured births) or on a bibliographic basis (references must be provided).

Checklist

The indication of the types is a very important contribution and the typification of some names gives extra value to the work done. Could you add a note in the checklist, when there are new typifications, in which you briefly explain why the choice was made? Also, can you comment on this aspect of the work in the Discussion?

 Considering that the work is based on a herbarium investigation, it would be opportune in the species list to add the herbaria acronyms of specimens where the species has been observed.

 There are no synonyms or incorrect reports in the list. There are none?

The abstract refers to discrepancies in the number of species present in Vietnam and to new species; could you illustrate and comment on these discrepancies and these new species in light of your results?

 Table

In the table I downloaded the rows are not aligned and therefore it is difficult to read.

In the manuscript, I commented on other minor points together with some suggestions.

Author Response

Reviewer 2

The suggested changes, additions and corrections were made:

R.:  The new species

In the Result section, there is the taxonomic treatment of the new species, which should be after the Discussion section because in the Result section you have to illustrate the systematics analysis, i.e. a comparison between the new population and other similar species in the genus, and after the Discussion section, where commenting the peculiarities of the new population (such as the Taxonomic remark included in the text), inserting the taxonomic treatment.

A.: The Taxonomic treatment of the new species was moved after the Discussion section. The comparison between the new population and the related species in the genus was illustrated in the Results section, after the Checklist of Capparis in Vietnam. In the Discussion section the peculiarities of the new population were commented.

R.: The risk status was assigned without giving detailed information on the surface occupied, the number and dispersion of the population(s) and the number of individuals.

A.: With regard to the risk status of the new species, information on the known locations in Hon Tre Island, and estimated number of individuals was given

R.: Some numbers are reported in the Results, particularly in the table. In Materials and Methods, it is necessary to explain what they indicate (mean, maximum, minimum, percentile, etc.) and on what basis they were obtained, whether on a statistical basis (number of populations, number of individuals and number of measured births) or on a bibliographic basis (references must be provided).

A.: In Materials and Methods it was indicated on what basis the morphometric data reported in the Description of the new species, and in Table 1 for the related taxa, were obtained. The data for  the new species were obtained by measurements of the herbarium material, and the range of variation of each character, from the minimum to the maximum value, was reported in the Description and Table. The data concerning the related species were obtained from bibliographic sources (quoted in Materials and Methods).

R.: Checklist

The indication of the types is a very important contribution and the typification of some names gives extra value to the work done. Could you add a note in the checklist, when there are new typifications, in which you briefly explain why the choice was made? Also, can you comment on this aspect of the work in the Discussion?

A.: In the Checklist notes concerning the lectotypification of three species were added, with reference to the relevant Articles of the International Code of Nomenclature for algae, fungi, and plants (Turland et al. 2018). Also in Discussion a comment on the species for which lectotypes were designated was added.

R.: Considering that the work is based on a herbarium investigation, it would be opportune in the species list to add the herbaria acronyms of specimens where the species has been observed.

A.: In the checklist, for each species were added the codes of the herbaria where the specimens were observed.

R.: There are no synonyms or incorrect reports in the list. There are none?

A.: For each taxon a list of synonyms was included in the Checklist.

R.: The abstract refers to discrepancies in the number of species present in Vietnam and to new species; could you illustrate and comment on these discrepancies and these new species in light of your results?

A.:  The discrepancies in the number of species present in Vietnam, as well as the species recently described, were commented in Discussion on the basis of our results.

R.: Table

In the table I downloaded the rows are not aligned and therefore it is difficult to read.

A.: In the table I tried to align the rows, probably the difficulty in reading it is due to the large size of the table. The final printout looks good.

R.: In the manuscript, I commented on other minor points together with some suggestions.

A.: Other changes suggested in the text were made. Among these a particular of Hon Tre Island was included in the map in figure 2.
